# Multiple signals evoked by unisensory stimulation converge onto cerebellar granule and Purkinje cells in mice

Misa Shimuta[1], Izumi Sugihara[2] & Taro Ishikawa [1✉]

The cerebellum receives signals directly from peripheral sensory systems and indirectly from the neocortex. Even a single tactile stimulus can activate both of these pathways. Here we report how these different types of signals are integrated in the cerebellar cortex. We used in vivo whole-cell recordings from granule cells and unit recordings from Purkinje cells in mice in which primary somatosensory cortex (S1) could be optogenetically inhibited. Tactile stimulation of the upper lip produced two-phase granule cell responses (with latencies of ~8 ms and 29 ms), for which only the late phase was S1 dependent. In Purkinje cells, complex spikes and the late phase of simple spikes were S1 dependent. These results indicate that individual granule cells combine convergent inputs from the periphery and neocortex and send their outputs to Purkinje cells, which then integrate those signals with climbing fiber signals from the neocortex.

---

[1] Department of Pharmacology, The Jikei University School of Medicine, Tokyo, Japan. [2] Department of Systems Neurophysiology, Tokyo Medical and Dental University Graduate School of Medical and Dental Sciences, Tokyo, Japan. ✉email: taroishi@gmail.com

The cerebellum and neocortex are interconnected by a large fiber system[1–3]. Although the basal ganglia similarly connect with the neocortex, the cerebellum receives inputs not only from the neocortex but also from the peripheral sensory systems, including tactile, proprioceptive, and vestibular systems[4]. How the cerebellum integrates these signals is not well understood.

The cerebellum receives inputs via two types of projection fibers, namely, mossy fibers that project to granule cells and climbing fibers that project to Purkinje cells[5]. A major subgroup of the mossy fibers projects from the pontine nuclei (basilar pontine nuclei and nucleus reticularis tegmenti pontis), which relay signals from the neocortex[3,6]. Other mossy fibers originate in the spinal cord and brainstem nuclei, including the trigeminal nuclei, and relay sensory signals directly from the periphery[7]. Beginning in the 1970s, it was shown that spinal and trigeminal mossy fibers transmitting somatosensory signals from a body part and the corticopontine mossy fibers transmitting signals from the corresponding somatotopic area in the somatosensory cortex terminate in the same areas in the cerebellum[6,8–12]. However, at the single-cell level, it is still not clear whether these inputs project to different groups of granule cells[9,12] or converge on the same individual granule cells. This issue is important in order to understand the basis of cerebellar computation.

Cerebellar granule cells are small electrically compact cells that receive synaptic inputs from a small number (on average, four) of mossy fibers. Conversely, a single mossy fiber projects to a much larger number (several hundred or more) of granule cells[5,13]. Thus, Marr and Albus independently proposed similar ideas that the mossy fiber-granule cell system expands the neural representation of information[14,15]. This idea, called the expansion recoding hypothesis, assumes that each granule cell receives inputs from a near-random combination of mossy fibers that convey different types of signals, thereby creating numerous combination patterns. Although experimental approaches to test this hypothesis have been hampered by technical difficulties, recent studies utilizing neuronal labeling and patch-clamp recording[16–18] have demonstrated that mossy fiber inputs conveying different sensory or motor signals converge onto single granule cells in some cases. However, as mentioned above, the fundamental issue of whether signals evoked by a particular sensory stimulus can be reunited at a granule cell after traveling via different pathways has not been investigated.

The cerebellar cortex also receives inputs from climbing fibers, which originate solely from the inferior olivary nuclei[5]. These nuclei comprise a large complex of subnuclei whose inputs have not fully been revealed. Although the inferior olivary nuclei receive indirect inputs from the neocortex[9,11,19–21], it is not known whether the neocortex mediates sensory-evoked climbing fiber responses.

In this study involving in vivo patch clamping and optogenetic manipulation, we provide functional evidence that direct trigeminal signals and indirect signals from the primary somatosensory cortex (S1) converge onto the same granule cells. We also show that this integration affects spike outputs of not only the granule cells but also the Purkinje cells. Furthermore, we show that the climbing fiber inputs to Purkinje cells also depend on activity in S1.

## Results

**The late component of cerebellar response is S1-dependent.** To investigate how inputs from S1 influence activity in the cerebellar cortex, we recorded field potentials simultaneously from S1 and the granule cell layer (GCL) of the cerebellar cortex in transgenic mice expressing channelrhodopsin 2 (ChR2) in GABAergic neurons (VGAT-ChR2 mice) (Fig. 1).

In mice anesthetized with ketamine/xylazine (K/X), a brief (50 ms) air puff applied to the upper lip evoked large responses both in the upper lip area of S1 (3.8–4.5 mm lateral and 0–1.0 mm rostral to bregma) and in the GCL of the crus II area. Similarly to that reported in rats[6], the responses in the GCL had two peaks, namely, early and late peaks ($8.3 \pm 0.3$ ms and $28.8 \pm 0.7$ ms [mean $\pm$ SEM] from the onset of stimulation, respectively, $n = 15$) that appeared before and after the peak of S1 ($25.2 \pm 1.2$ ms, $n = 15$), respectively (Fig. 2a, b). When S1 activity was suppressed by focal illumination with blue light in alternating trials, the response of S1 and the late (but not early) component of the GCL response were eliminated (Fig. 2a, b), indicating that the late component of the GCL response depends on the activity of S1. For this and subsequent experiments, a continuous 150 ms light stimulus was applied during sensory stimulation, but a longer or shorter illumination covering the timing of the sensory stimulation was similarly effective (Supplementary Fig. 1a–e). In similar experiments performed in awake head-fixed mice, the S1 response had a peak at $14.2 \pm 1.0$ ms ($n = 9$) and the GCL response had three peaks. The first and the second peaks of GCL (at $5.1 \pm 0.2$ ms and $13.0 \pm 0.3$ ms from the onset of stimulation, respectively, $n = 9$) were not affected by photoinhibition of S1 (Fig. 2c, d), whereas the third peak (at $21.4 \pm 0.8$ ms from the onset of stimulation, $n = 9$) was suppressed by photoinhibition of S1 (Fig. 2c, d). Taking their timings and S1 dependencies into account, these results suggested that the first two peaks were equivalent to the early component in the K/X anesthetized mice, whereas the third peak was equivalent to the late component. Of note, the photosensitive component was larger in K/X anesthetized mice than in awake mice (Fig. 2e), presumably because K/X generates synchronized neocortical activity, which resembles that

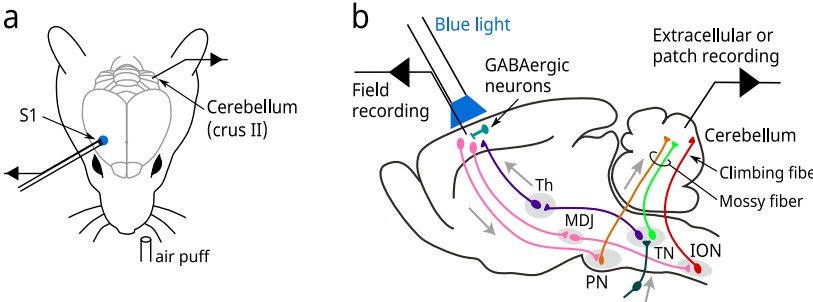

**Fig. 1 Experimental configuration and neural circuit involved in this study. a** Tactile stimulation (air puff) was applied to the left upper lip while field potentials were recorded from the GCL of the ipsilateral crus II area of the cerebellum and the upper lip area of the contralateral S1. S1 was optogenetically suppressed in alternating trials. **b** Schematic of the neural circuit involved in this study. Th thalamus, MDJ meso-diencephalic junction, PN pontine nuclei, TN trigeminal nuclei, ION inferior olivary nucleus.

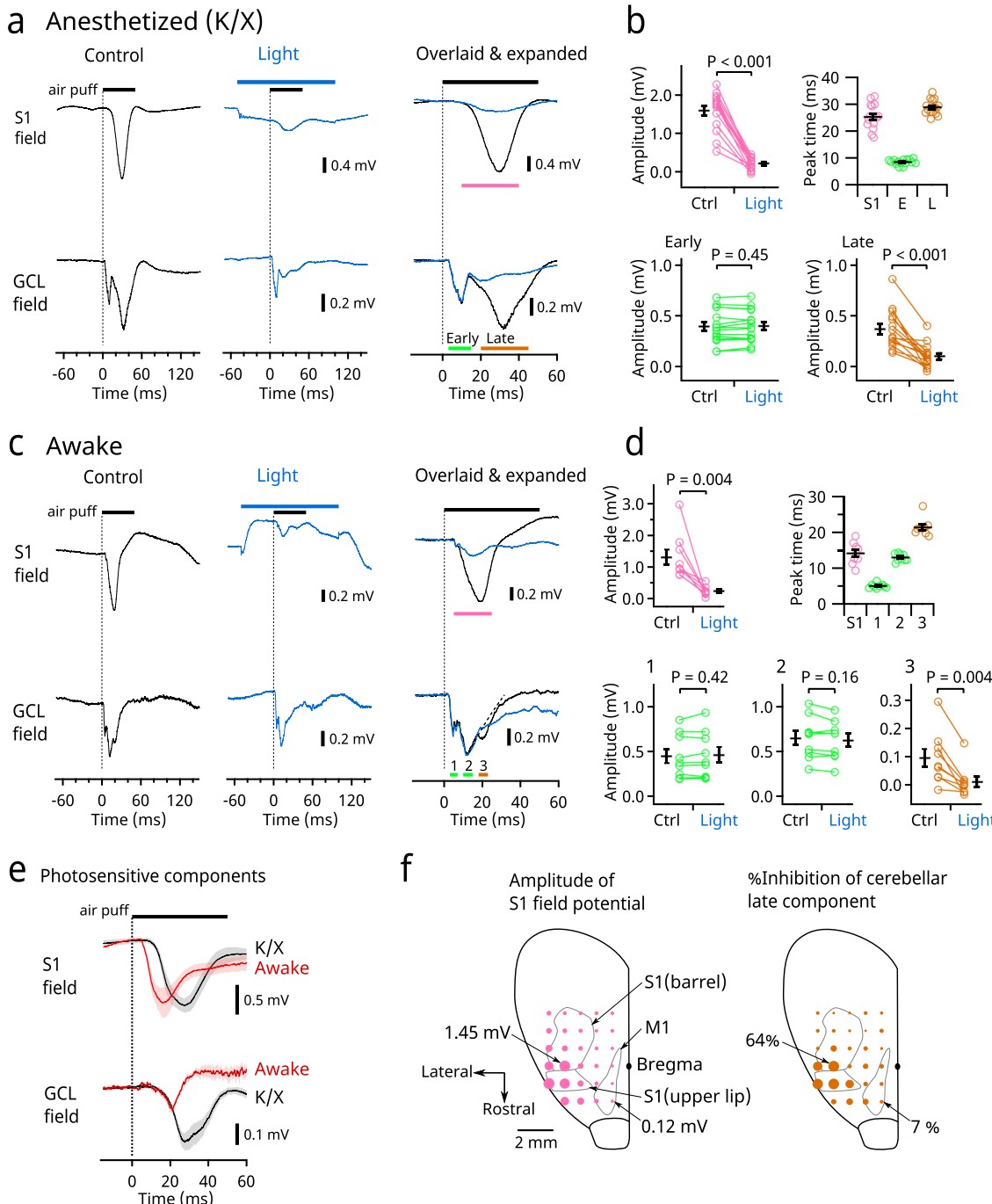

**Fig. 2 Optogenetic inhibition of S1 suppresses the late component of the cerebellar sensory response in field potential recordings. a** Representative recordings from a mouse anesthetized with ketamine/xylazine (K/X); 20 traces for each condition were averaged. The same traces are baseline adjusted and overlaid in an expanded time scale in the panels on the right. Black and blue bars indicate the durations of air puff and LED illumination, respectively. The vertical dotted lines indicate the onset of air puff. **b** Pooled data from 15 anesthetized mice. Peak amplitudes were measured during the times indicated by colored bars in **a**. The top right panel shows timing of peaks. **c** Similar to **a** except using an awake mouse; 46 traces for each condition were averaged. The decay time course of the second component of the cerebellar response was fitted and extrapolated by a single exponential curve (the dashed line) in the panel on the right. **d** Similar to **b** except using nine awake mice. The cerebellar late component was measured after subtracting the decay component of the early response (see "Methods" section). **e** Photosensitive components obtained as differences between the control and the light conditions in K/X ($n = 15$) and awake ($n = 9$) mice. Shading indicates SEMs. **f** Left, bubble size is proportional to the field potential amplitude at each spot measured in the right cerebral hemisphere. Right, bubble size is proportional to the magnitude of inhibition of the cerebellar late component when blue light was applied to each spot. These two values significantly correlated ($P < 0.001$, $n = 29$ spots). The cerebellar early component was not affected by photoinhibition at any of these spots. Data from 15 animals under K/X were combined. Each spot is the average from 3–15 animals. S1 for upper lip and barrel and the primary motor cortex (M1) are illustrated in accordance with Allen Mouse Brain Atlas and Mohajerani et al.[72]. Means ± SEMs are presented as black bars and lines, respectively.

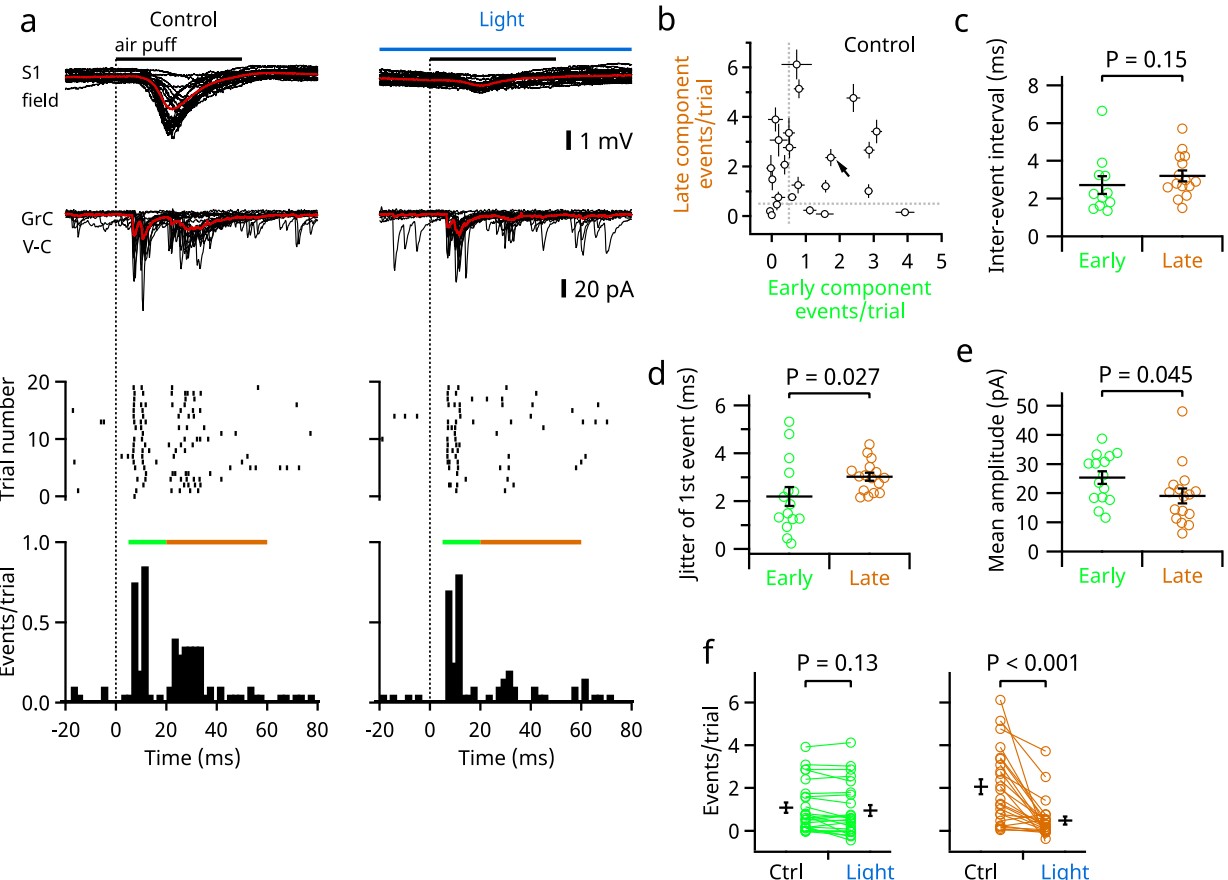

**Fig. 3 In vivo whole-cell recording from granule cells revealed convergent synaptic inputs. a** Representative recordings; simultaneous field potential recording from S1 (top) and whole-cell voltage clamp recordings from granule cell (lower); 20 consecutive traces are overlaid. The averaged traces are in red. Detected EPSC events are shown in raster plots (middle) and time histograms (bottom). Green and brown bars indicate the early and the late phases, respectively. Trials under control (left) and the light (right) conditions were interleaved. **b** Numbers of evoked EPSC events in the early and the late phases were compared in 24 cells. The numbers were corrected for baseline spontaneous events by subtraction in this and the following panels. For qualitative description, event numbers of <0.5 events/trial (dotted lines) were defined as no response. Error bars indicate SEMs (trial-by-trial fluctuation). The arrow points to the cell recorded in **a**. **c** Median interevent intervals of each cell were compared ($n = 11$, early; $n = 15$, late). **d** Trial-by-trail fluctuations (measured as standard deviations) of the timing of the first event during each time period were compared ($n = 15$, early; $n = 16$, late). **e** Mean amplitudes of individual EPSCs in the early and late phases were compared ($n = 15$, early; $n = 16$, late). **f** EPSC event numbers were compared for the early phase (left) and the late phase (right) ($n = 24$). Means ± SEMs are presented as black lines and bars, respectively.

in natural slow-wave sleep[22–24] (see "Discussion" section). In a separate set of experiments, anesthetics other than K/X had suppressive effects on both the S1 response and the late component of the GCL response (Supplementary Fig. 1f, g). Therefore, we used K/X in subsequent experiments.

To identify the neocortical region involved in the cerebellar response, we performed the same experimental protocol but with the optrode at various loci of the neocortex. The late component of the GCL response was suppressed most effectively by photoinhibition of the locus where the S1 response was largest (3.8 mm lateral and 0 mm rostrocaudal from bregma) but not suppressed by photoinhibition at distant loci, including the primary motor cortex (M1) (Fig. 2f). This result suggested that S1 projects directly to the pons, without a relay through other neocortical areas.

**Convergent synaptic inputs to single granule cells.** The above-described results were consistent with the established theory that at least two distinct groups of mossy fibers project to the GCL of crus II: one directly from the trigeminal nuclei and the other via the cerebropontine pathway[6,25,26]. To determine whether these two inputs converge onto individual granule cells, we performed whole-

cell patch-clamp recordings in anesthetized mice. Granule cells in the central part of crus II (corresponding to 5−, 6+, and 6− bands) were sampled without visual inspection. In 50% (12/24) of the granule cells, excitatory postsynaptic currents (EPSCs) evoked by stimulation of the upper lip had two components (as in the representative cell in Fig. 3a), indicating that two types of mossy fibers converge on some individual cells. However, other granule cells had conspicuous EPSCs with only early (12.5%) or late (25%) timing, and the remainder (12.5%) had no response (cutoff, 0.5 events/trial) (Fig. 3b). The numbers of EPSC events for the two components did not correlate (Pearson's linear correlation coefficient, $r = 0.028$, $n = 24$, $P = 0.90$), suggesting that these connections were established independently. In these recordings, it was noted that each component often had multiple EPSC events with very short intervals (around 3 ms on average; Fig. 3c). This is in line with previous studies showing that a single mossy fiber can fire high-frequency bursts of action potentials that trigger high-frequency EPSCs in granule cells[27,28]. Indeed, in our occasional whole-cell recordings from putative mossy fiber boutons ($n = 3$), action potentials occurred in high-frequency bursts with either early or late timing (Supplementary Fig. 2). Thus, the multiple EPSCs in each component were likely derived from a single mossy fiber. Furthermore, as the interevent intervals for the early and the

late EPSC components did not differ (Fig. 3c), it is likely that tri-geminal and pontine mossy fibers can fire similar high-frequency bursts. However, the fluctuation of the timings (jitter) of the first event was larger in the late component than in the early component (Fig. 3d), reflecting a longer multistep pathway for transmission of the late response. However, the amplitude of individual EPSC events was larger for the early components (Fig. 3e), suggesting that the synaptic properties (i.e., quantal content and/or quantal size) may differ between these two types of synaptic inputs, as reported pre-viously for the various vestibular inputs[16]. As expected, the late component of EPSCs was mostly eliminated by optogenetic sup-pression of S1 (Fig. 3f), confirming that the late component signal is derived from S1. Similar results were obtained when synaptic charge (the area over the curve), instead of event number, was measured as an indicator of synaptic strength (Supplementary Fig. 3a, b). Fur-thermore, spontaneous EPSCs were partially blocked by the sup-pression of S1 (Supplementary Fig. 3f, g). These results indicate that granule cells receive convergent inputs from two types of mossy fibers, although the balance of these inputs varies between cells. In addition, we recorded inhibitory postsynaptic currents (IPSCs) in granule cells ($n = 6$) to examine feed-forward inhibition from Golgi cells. IPSCs also exhibited two components, and the late component was suppressed by light illumination of S1 (Supplementary Fig. 3c–e).

**Spike output of single granule cells**. To investigate how synaptic inputs trigger action potentials in granule cells, we recorded membrane potentials in current clamp (Fig. 4a). The resting potentials of granule cells varied widely (from $-96.0$ to $-50.8$ mV; mean, $-72.9 \pm 3.4$ mV, $n = 16$). In cells that had a relatively hyperpolarized resting potential, the excitatory postsynaptic potential (EPSP) did not reach the spike threshold (Fig. 4b), indicating that a single tactile stimulus generated a spike in only a subset of the granule cells. As the resting potentials of granule cells are modulated by multiple factors, such as tonic inhibition from Golgi cells and potassium channel function[29–31], we applied steady depolarizing current (up to 20 pA) to 9 of the 16 cells, thereby raising the average resting potential to $-57.2 \pm 1.9$ mV ($n = 16$, including seven cells without adjustment) and increasing their propensity to fire in response to tactile stimulation of the upper lip (11/16 cells fired action potentials; cutoff, 0.5 spikes/trial) (Fig. 4a, b). Under this condition, the granule cells fired during the early or late phase (Fig. 4c). As expected, action potentials in the late phase were eliminated by photoinhibition of S1 (Fig. 4f). Investigation of the relationship between synaptic inputs and firing outputs showed that the number of action potentials evoked in the early phase correlated with the number of EPSC events during the same period ($r = 0.77$, $n = 13$, $P = 0.002$) (Fig. 4d, left). However, the correlation was less clear in the late phase ($r = 0.53$, $n = 13$, $P = 0.063$) (Fig. 4d, right), suggesting that factors other than instantaneous synaptic inputs may be involved. Indeed, depolarization caused in the early phase was maintained in the late phase (Fig. 4e). This suggests that action potential firing in the late phase may be facilitated by temporal summation. Although we could not directly extract such a facil-itating effect from our experimental data on the number of action potentials because of the involvement of many other factors (including the probability of firing in the early phase and the size of synaptic inputs in the late phase), a simulation with a realistic computational model of a granule cell demonstrated facilitation of firing in the late phase by temporal summation even when counteracting factors were included in the model[32] (Supple-mentary Fig. 4).

**Granule cell responses in various parasagittal bands**. The cer-ebellar cortex is organized in parasagittal bands defined by the

expression of aldolase C. In the above-described experiments, we positioned the cerebellar recording electrodes coarsely around the 6+ band in crus II. As recent morphological studies indicated that corticopontine mossy fibers project to aldolase C-positive bands[33,34], we set out to characterize the distribution of early and late responses in crus II by using Aldoc-Venus mice in which the various bands can be visualized (Fig. 5a, b). In vivo whole-cell voltage clamp recordings were made from the cerebellar granule cells in 5+, 5−, 6+, and 7+ bands simultaneously with field potential recordings from S1. As observed in VGAT-ChR2 mice, some granule cells had EPSCs in both early and late time periods, whereas others had EPSCs in only one of these time periods (Fig. 5a). Early and late responses were detected in all bands, and the numbers of early EPSC events were similar across all bands ($P = 0.95$, see "Methods" section for details of statistical tests) (Fig. 5c). However, the numbers of late EPSC events were sig-nificantly higher in the 7+ band than in the 5− band ($P = 0.005$, pairwise multiple comparisons; the other five pairs had no sig-nificant difference) (Fig. 5c). In terms of the balance between the early and the late components, consistent results were obtained with field potential recordings in the GCL (Supplementary Fig. 5a, b). These results suggest that the putative trigeminal and corticopontine mossy fibers are distributed differently over the parasagittal bands in crus II, although their innervation was not entirely selective to particular bands. Furthermore, trial-by-trial analysis revealed a correlation between the amplitude of the S1 response and the number of EPSC events in the late component but not in the early component (Fig. 5d, e), confirming that only the late component is S1 dependent. Similar to that observed in VGAT-ChR2 mice, the mean EPSC amplitude was larger and the jitter of the first event was smaller in the early response than in the late response, and the interevent intervals did not differ (Supplementary Fig. 5c–e).

**Effect of S1 photoinhibition on SS and CS in Purkinje cells**. We next investigated how the suppression of S1 affects the firing of Purkinje cells, which are downstream of granule cells in the cerebellar circuit. In general, simple spikes (SS) in Purkinje cells are generated spontaneously but are affected by excitatory synaptic inputs from parallel fibers (i.e., granule cell axons) and inhibitory synaptic inputs from molecular layer interneurons, whereas complex spikes (CS) are triggered solely by climbing fiber inputs. Under control conditions (without S1 inhibition), there were four phases of SS in response to stimulation of the upper lips of VGAT-ChR2 mice, namely, early excitatory (within 10 ms after stimulation onset), early inhibitory (around 10–20 ms), late excitatory (around 20–30 ms), and late inhibitory (30–60 ms) phases (Fig. 6a–d). CS coincided with the late inhibitory phase of SS. Photoinhibition of S1 did not affect the early phases of SS but eliminated the late excitatory and inhibitory phases (Fig. 6a–d). Interestingly, the CS response was also abolished. These results suggest that the direct trigeminal inputs to granule cells trigger the early excitatory and inhibitory phases of SS, the latter of which presumably results from the activation of molecular layer interneurons[35,36]. These results also suggest that the late excita-tory responses of SS reflect inputs from S1. However, the late inhibitory response could reflect either an interneuronal effect or an intrinsic pause after CS. As CS did not always occur under the control condition, we were able to separate traces with CS from those without (Supplementary Fig. 6a, b). The late inhibition phase of SS was larger in traces with CS than in those without CS, suggesting that the late inhibition largely reflects the intrinsic pause after CS. We also tested the effects of photoinhibition at multiple loci in the neocortex and found that suppression of the upper lip area of the contralateral S1 was most effective in

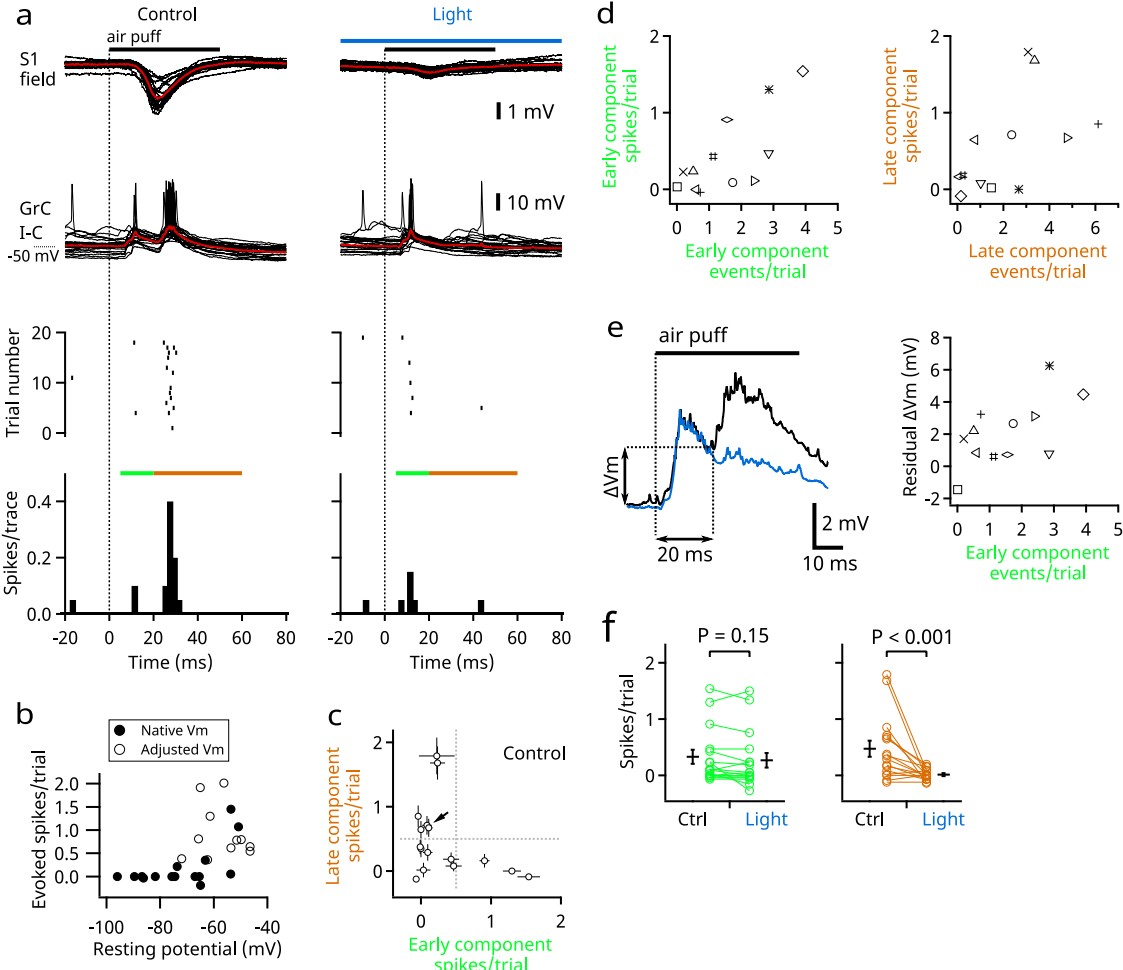

**Fig. 4 Spike output of single granule cells in vivo. a** Representative recordings of the same cell as in Fig. 3a; simultaneous field potential recordings from S1 (top) and whole-cell current clamp recordings from a granule cell (lower); 20 traces are overlaid. The averaged traces are in red. Detected spikes are shown in raster plots (middle) and time histograms (bottom). Green and brown bars indicate the early and late phases, respectively. The trials under control (left) and light (right) conditions were interleaved. **b** Numbers of evoked spikes plotted against the native resting potentials (filled circles, $n = 16$). In a subset of cells that had hyperpolarized native resting potentials ($<-65$ mV), a steady depolarizing current was applied to facilitate firing (open circles, $n = 9$). The spike numbers were corrected for baseline spontaneous firing by subtraction in this and the following panels. **c** Numbers of evoked spikes in the early and the late phases were compared in 16 cells, including three cells that lacked voltage clamp recordings. Error bars indicate SEMs (trial-by-trial fluctuation). The arrow points to the cell recorded in **a**. **d** Numbers of evoked spikes in the early phase (left) and late phase (right) were plotted against the numbers of EPSCs during the same time periods. Individual cells ($n = 13$) are plotted as different marks common in **d** and **e**. **e** Left, averaged traces of all 16 granule cells in current clamp mode. Black, control; blue, photoinhibition of S1. Right, depolarization at 20 ms after the onset of stimulation is plotted against the number of EPSCs in the early phase. **f** Evoked spike numbers were compared for the early phase (left) and the late phase (right) ($n = 16$). Means ± SEMs are presented as black lines and bars, respectively.

blocking CS in Purkinje cells (Fig. 6e). Furthermore, suppression of S1 with a long (10 s) light stimulus inhibited spontaneous CS firing and triggered rebound activation after the light was turned off, suggesting that the spontaneous activity of the inferior olive is under strong control of S1. However, this photoinhibition had no effect on SS firing (Supplementary Fig. 6c, d), suggesting that S1 activity is not directly linked to the spontaneous activity of granule cells (Supplementary Fig. 3f, g), in line with a previous report[20].

## Discussion
By taking advantage of the high temporal resolution of electrophysiology and optogenetics, we obtained functional evidence that cerebellar granule cells in crus II receive inputs directly from the periphery and indirectly via S1. We found that approximately half of the granule cells receive convergent signals from both

pathways. We also showed that the olivocerebellar inputs to these Purkinje cells come through the same area in S1.

We adopted an optogenetic method in which photostimulation of GABAergic neurons expressing ChR2 can suppress the activity of specific areas of the neocortex[37,38]. This method is based on the fact that most GABAergic neurons in the neocortex are local interneurons. Although 0.5% of GABAergic neurons project to other cortical and subcortical areas[39], it is unlikely that such long-range GABA projections contributed to our findings because light illumination was effective only in specific areas in the neocortex (Figs. 2f and 6e). Moreover, there is anatomical and physiological evidence indicating that the basal pons relays signals from S1 to the cerebellar cortex[1,3,6,40]. Therefore, it is highly likely that the late component of the cerebellar granule cell response in our study was transmitted via the cortico-ponto-cerebellar pathway.

Recent experimental evidence[16–18] supports the conventional view of Marr and Albus[14,15] that granule cells receive inputs from

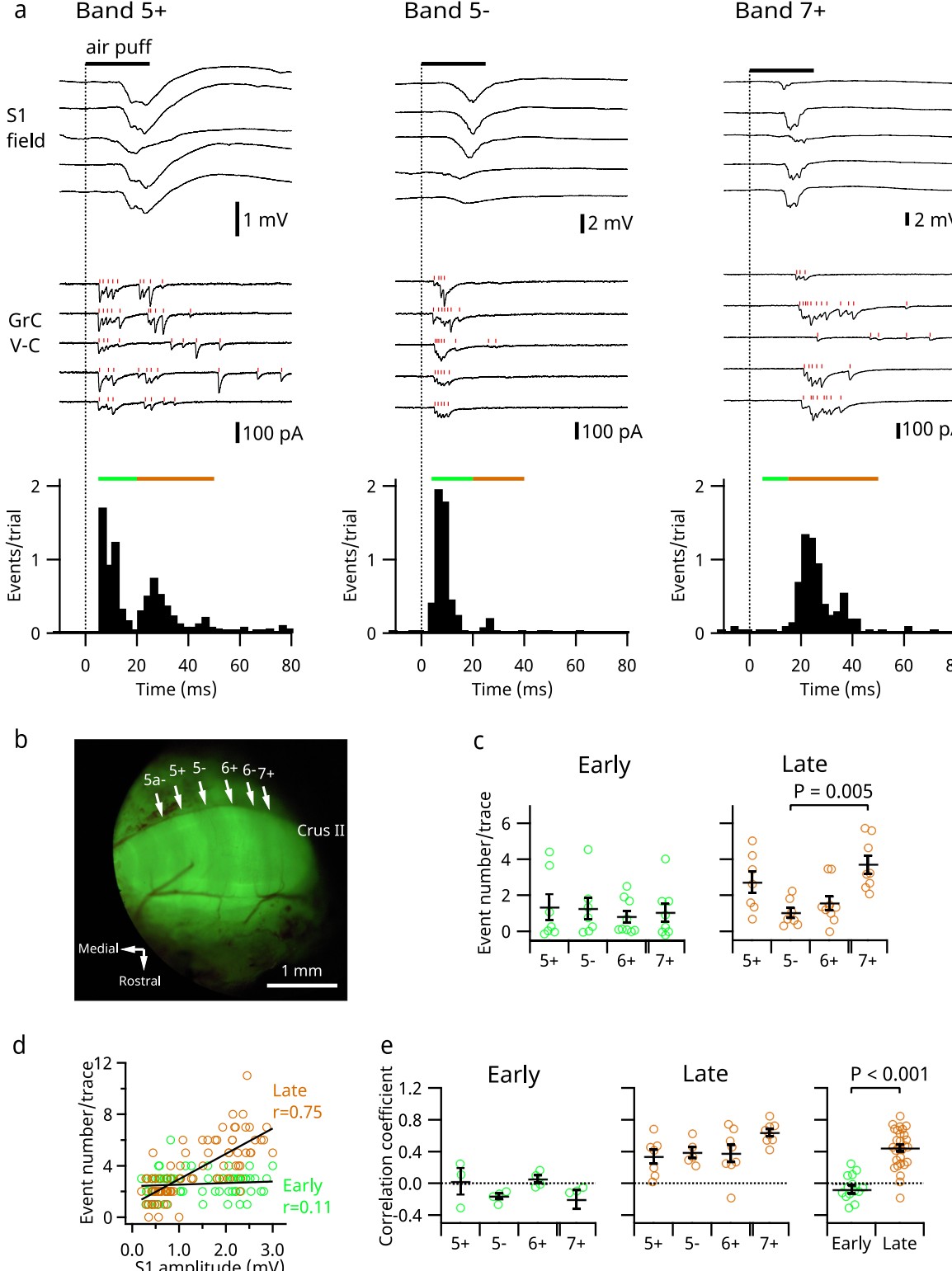

**Fig. 5 Sensory-evoked responses in granule cells in various parasagittal bands. a** Representative recordings; simultaneous field potential recordings from S1 (top) and whole-cell voltage clamp recordings from a granule cell (lower) in Aldoc-Venus mice; five consecutive trials from each cell are displayed. Overlaid raster plots in red indicate the timing of detected EPSC events. Time histograms are shown in the bottom panels. Green and brown bars indicate the early and the late phases, respectively. **b** Fluorescent macroscopic image of the cerebellar crus II area after removal of the dura mater. **c** Numbers of EPSC events in different bands in the early (left) and the late (right) phases ($n = 7$, 7, 9, and 8 for 5+, 5−, 6+, and 7+, respectively). **d** Data from a sample cell in the 6+ band. The EPSC event numbers in the early (green) and late (brown) phases in every trial were plotted against the amplitudes of S1 field responses. Correlation coefficients were calculated for each phase. **e** Correlation coefficients of all cells are plotted. The right panel shows pooled data from all bands. Means ± SEMs are presented as black lines and bars, respectively.

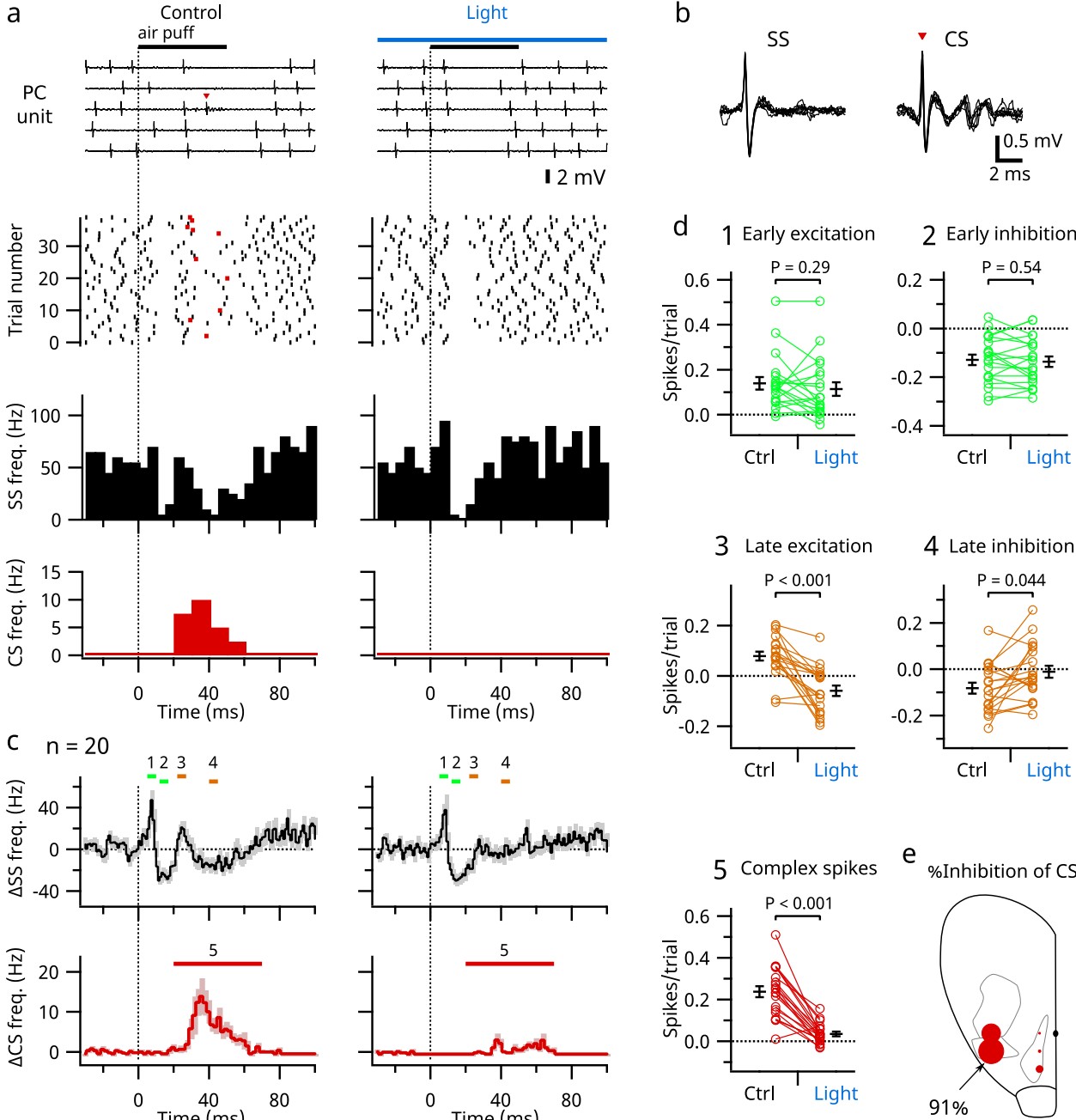

**Fig. 6 Effects of S1 photoinhibition on SS and CS in Purkinje cells. a** Representative extracellular unit recordings of a Purkinje cell; five traces are displayed (top). Simultaneous field potential recordings from S1 are provided in Supplementary Data 1. SS (black) and CS (red) are plotted in the raster plots (lower). Separate time histograms of SS (middle) and CS (bottom) from 40 trials of each condition are shown. **b** Enlarged views of SS and CS of the unit shown in **a**; six traces in each were aligned and overlaid. **c** Averaged time histograms of SS (1 ms bin) and CS (2 ms bin) from 20 Purkinje cells. Shading indicates SEMs. Vertical dotted lines indicate the onset of stimulation as in **a**. **d** Spike numbers were measured during the times (1–5) indicated by colored bars in **c**. Means ± SEMs are presented as black lines and bars, respectively. **e** Bubble size indicates magnitude of inhibition of the cerebellar CS when the blue light was applied to each spot in the right cerebral hemisphere. The magnitude of inhibition significantly correlated with the size of field potential at each spot ($P = 0.011$, $n = 5$ spots). Data from six animals were combined. Each spot is the average from 3–6 animals.

mossy fibers carrying different types of information. However, it is also proposed that individual granule cells receive only one type of input[41–43]. In our present study, although both trigeminal and corticopontine signals were evoked by the same tactile stimulation, these signals are fundamentally different because they can be differentially modulated on route to the cerebellum. For instance, neocortical states (such as anesthesia and wakefulness) affected them differently (Fig. 2), and they exhibited different trial-to-trial fluctuations (Fig. 5). Furthermore, as the S1 in mice sends efferent

motor signals directly to the brain stem[44], the corticopontine signals we observed may in fact be efferent copies of a motor command. Thus, as the distinctions between trigeminal and corticopontine signals are substantial, our results are in line with the conventional idea proposed by Marr and Albus.

However, since granule cells received either or both types of inputs, it is difficult to draw a unified view for these connections. For the subpopulation of granule cells that receive both, our results (Fig. 4e) indicate that depolarization during the early

phase may facilitate firing in the late phase. In such cases, the granule cells represent coincident detectors (i.e., an "AND" gate) and pattern recoders as proposed by Marr and Albus. By contrast, granule cells that fire in response to a burst of synaptic input from either a direct or S1-mediated connection may work as a simple relay or a frequency filter[28,41,42]. Given that S1 can be activated by top-down control[45,46], we speculate that a granule cell can become an "OR" gate if both direct and S1-mediated inputs are strong enough to trigger action potentials independently. An experimental system in which trigeminal and corticopontine pathways can be activated independently is required to test this.

Previous morphological studies showed that aldolase C-positive bands have more corticopontine inputs than trigeminal or spinal ones and that aldolase C-negative bands have the opposite trend[33,34], although there may be some overlap[47]. Our present results (Fig. 5 and Supplementary Fig. 5) are consistent with those studies, indicating that different types of mossy fibers have some tendency to innervate different compartments. However, at the single-cell level, the strengths of these two types of inputs were neither positively nor negatively correlated but had diverse patterns of connections (Fig. 3b), suggesting that two independent mossy fibers can synapse with a granule cell without attracting or repelling each other[48]. The diversity of these connections is consistent with the expansion recoding hypothesis of granule cells proposed by Marr and Albus[13].

Studies since the 1970s have proposed that different types of mossy fibers project to distinct depths in the GCL[12,49,50]. A recent study using extracellular recordings suggested that the trigeminal mossy fibers are located deeper than the corticopontine mossy fibers[12]. However, since we found that many granule cells receive both types of inputs, the trigeminal and corticopontine mossy fibers should largely overlap in the GCL even if they tend to be distributed at different depths. This idea is consistent with the fact that individual mossy fibers branch to widely different depths[51]. Future studies, such as those using functional cellular imaging[52,53], may elucidate the detailed organization of the GCL.

In line with the established notion that Golgi cells integrate mossy fiber inputs and give feed-forward inhibition to granule cells[29,54–56], we found that sensory-evoked IPSCs in granule cells had two phases similar to those of EPSCs (Supplementary Fig. 3a–e). The IPSCs were delayed in relation to EPSCs by only a few milliseconds, reflecting rapid feed-forward inhibition from Golgi cells. The importance of the excitation/inhibition balance in granule cells has been discussed elsewhere[57,58].

We found that Purkinje cells receive signals from the neocortex not only via the mossy fiber-parallel fiber pathway but also via climbing fibers. This finding is consistent with anatomical observations that Purkinje cells in the 6+ compartment (D1 band) receive inputs from climbing fibers from the ventral principal olive, which receives inputs from the neocortex most likely via the area parafascicularis prerubralis in the meso-diencephalic junction[59–61]. Our present study is also in line with classical physiological studies showing a convergence of mossy and climbing fiber signals originating from the same neocortical areas in cats and monkeys[9,11,21], as well as with a more recent study showing that injections of lidocaine into the somatosensory cortex delay the timing of CS in crus II in rats[19]. Still, our present study is the first, to our knowledge, to directly demonstrate that S1 mediates sensory-evoked climbing fiber responses. In contrast to what we observed, a recent study by Kubo et al.[62] reported that inhibition of the neocortex in mice did not inhibit CS. The reason for this discrepancy is unclear, but it may be that they explored only medial parts of the neocortex.

The physiological significance of this cortio-olivo-cerebellar connection is intriguing, especially given the accompanying cortico-ponto-cerebellar connection. This circuitry suggests that the neocortex exerts powerful control over cerebellar signal processing. For instance, it may be possible that signals from the neocortex induce synaptic plasticity at synapses on Purkinje cells when parallel and climbing fibers are activated simultaneously via the respective cortico-ponto-cerebellar and cortio-olivo-cerebellar pathways. Such top-down signaling may occur not only during wakefulness but also during sleep as offline activity[46].

One limitation of this study is that the data were largely obtained from animals anesthetized with K/X, which has a sup-pressive effect on parallel fiber-Purkinje cell synapses[29,63]. How-ever, the sensory stimulation applied was still able to evoke SS in Purkinje cells under K/X (Fig. 6). More importantly, K/X syn-chronizes the activity of the neocortex, rendering a condition similar to natural slow-wave sleep[22–24]. Considering this effect and the fact that S1-mediated signals in the cerebellum were larger in anesthetized animals than in awake animals (Fig. 2), it will be interesting to see whether those signals are enhanced in slow-wave sleep.

A second limitation is that we did not have a method to spe-cifically block the trigemino-cerebellar pathway. A future chal-lenge is to manipulate the trigeminal neurons projecting to the cerebellum without affecting those projecting to the thalamus. This may be possible, in principle, as those neurons may be different populations in the trigeminal nuclei[64,65]. On a related note, we do not know whether the early phase inputs to the cerebellar GCL, including the second peak in awake mice, were monosynaptic or polysynaptic. It is possible that rapid poly-synaptic inputs contributed, such as those relayed within the trigeminal nuclei or in other brain areas, including the pontine nuclei, which send mossy fibers to the cerebellar cortex[66], and the cerebellar nuclei, which project via nucleo-cortical mossy fibers[67].

Finally, in this study, the only sensory stimulus was tactile stimulation (air puff) of the upper lip. We adopted this form of stimulation because it gave large responses in both S1 and the cerebellum. However, tactile stimulation of other body parts or stimulation of other modalities may similarly evoke multipathway responses in the cerebellar cortex. Given the extensive connec-tions between the neocortex and the cerebellum[2,3], it is likely that similar circuits exist for the sensation of other types of stimulation.

We found that sensory-evoked signals directly from the per-iphery and indirectly from the neocortex are integrated in the GCL, the first stage of cerebellar processing, and that results of this integration are reflected in the firing of Purkinje cells. As the firing of Purkinje cells directly affects that of neurons in the deep cerebellar nuclei in precise timings[68], our findings suggest that the interactions of multiple types of mossy fiber inputs have a sub-stantial impact on the output of the cerebellar circuit. Moreover, it can be speculated that the cerebellar circuit can be modulated by synaptic plasticity induced by neocortex-mediated climbing fiber activity.

## Methods

**Animals**. All animal procedures were approved by the Institutional Animal Care and Use Committee of The Jikei University (no. 2015-054 and 2017-001) under the Guidelines for the Proper Conduct of Animal Experiments of the Science Council of Japan (2006). Hemizygous transgenic mice (3–8 weeks old, male and female) expressing modified ChR2 (hChR2-H134R) fused with YFP in GABAergic neurons via the vesi-cular GABA transporter promoter/enhancer (VGAT-ChR2 mice, stock no. 014548; Jackson Laboratory)[69] were used in photoinhibition experiments. Heterozygous knock-in mice (3–5 weeks old, male and female) expressing Venus fluorescent protein via the aldolase C promoter (Aldoc-Venus mice, MGI:3609644)[70] were used to visually identify the cerebellar parasagittal zones.

**Surgery**. The mice were anesthetized with an initial dose of a mixture of ketamine (86 mg/kg body weight) and xylazine (10 mg/kg), and supplemented with a con-tinuous infusion of ketamine (70 mg/kg/h) and xylazine (8 mg/kg/h) via a syringe pump to maintain a stable level of anesthesia with free breathing. The heads of

mice were fixed in position via a head post glued onto the skull. Core body temperature was maintained at around 37 °C with an isothermic feedback heating pad. After removing the overlaying skin and muscles, a craniotomy was performed over the left cerebellar crus II area and the right cerebral somatosensory area. After removing the dura, the exposed brain surface was kept moist with a HEPES-buffered saline containing (in mM) NaCl (150), KCl (2.5), HEPES (10), CaCl$_2$ (2), and MgCl$_2$ (1) (pH adjusted to 7.4 with NaOH). To record from awake mice, the head post was attached during a surgery 3 days before recording. Craniotomies were performed under isoflurane anesthesia on the recording day. The mice were habituated to the recording environment for >1 h before the start of recording. The duration of the recording session was <5 h for each animal.

**Recording**. Whole-cell in vivo patch-clamp recordings were performed via a resistance-guided (blind) method as previously described[18,28]. Voltage clamp and current clamp recordings were made from granule cells at a depth of 200–400 μm in crus II of the cerebellar cortex by using a Multiclamp 700B amplifier (Molecular Devices). Data were low-pass filtered at 6 kHz and acquired at 50 kHz using a USB-6259 interface (National Instruments) and Igor Pro with NIDAQ Tools MX (WaveMetrics). The internal solution contained (in mM) K-methanesulfonate (135), KCl (7), HEPES (10), Mg-ATP (2), Na$_2$ATP (2), Na$_2$GTP (0.5), and EGTA (0.05 or 0.1) (pH adjusted to 7.2 with KOH), giving an estimated chloride reversal potential of −69 mV. This enabled excitatory synaptic currents to be observed in isolation by voltage clamping at −70 mV and inhibitory synaptic currents to be observed at 0 mV. The liquid junction potential was not corrected. In a subset of experiments, biocytin 0.5% was added to the internal solution. The granule cells were identified by a small membrane capacitance (C$_m$) (<7 pF), a high membrane resistance (>0.5 GΩ; on average 1.57 ± 0.16 GΩ, $n = 25$), and a lack of periodic spontaneous firings. The mossy fiber boutons were identified by the occurrence of high-frequency bursts and a lack of synaptic potentials[28]. Patch pipettes had resistances of 5–8 MΩ, and series resistances (R$_s$) were typically 30–50 MΩ. Cells with high R$_s$ were excluded to keep the access time constant (R$_s$·C$_m$) at <0.3 ms. Extracellular unit recordings from Purkinje cells were made with glass electrodes (5 MΩ) filled with saline. The Purkinje cells were identified by the occurrence of SS and CS. Field potential recordings were conducted using a tungsten electrode (1 MΩ, TM31A10; WPI). Neocortical field potentials were recorded at a depth of 500–600 μm in the primary somatosensory area for the upper lip (3.8 mm lateral and 0 mm rostrocaudal from bregma) unless otherwise noted. Cerebellar field potentials were recorded in the GCL at a depth of 300–500 μm in the crus II area. In VGAT-ChR2 mice, the somatosensory cortex was optogenetically inactivated by illuminating the surface with an optic fiber (0.39 NA, Ø400 μm, T400EMT; Thorlabs) coupled to a high-power blue LED (470 nm, M470D2; Thorlabs). The optic fiber was bundled with a single tungsten electrode, and the tip of the electrode was 800 μm ahead of that of the optic fiber so that the tip of the optic fiber was above the surface of the brain during the recording. Note that in the experiments for Fig. 2f, 29 spots separated mediolaterally by 0.9 mm steps and rostrocaudally by 1.0 mm steps from bregma were tested. In the experiments for Fig. 6e, only five spots on the same grid were tested. To record from visually identified bands in Aldoc-Venus mice, the angle of electrode was carefully adjusted to be perpendicular to the brain surface, and the center of each band was targeted. However, due to the limited spatial specificity inherent in field potential recording[71], contamination of field potentials from neighboring bands was not fully excluded. For whole-cell recordings from those mice, only 5+, 5−, 6+, and 7+ bands, which were relatively wide and easy to identify, were used. Tactile stimulation (air puff, 50 ms, 90 mmHg at the source) was applied to the left upper lip with a solenoid valve (FAB31-8-3-12C-1; CKD, Aichi, Japan) controlled via the USB-6259 interface as described above.

**Analysis**. To analyze field potential recordings, the peak amplitude from the baseline was measured. The early component was measured 3 to 15 ms from the onset of stimulation, but subdivisions of this period (3–7 ms and 10–15 ms) were used for awake mice. In K/X-anesthetized mice, the baseline for the cerebellar late component was set at the trough between the early and late components, which was around 15 ms from the onset of stimulation. In awake mice and mice anesthetized with a mixture of medetomidine (0.3 mg/kg), midazolam (4 mg/kg), and butorphanol (5 mg/kg) or with isoflurane, the late component was measured as the mean amplitude (i.e., the average of all points in the time window) 18–23 ms (fixed across all cases) from the stimulation onset after subtracting the extrapolated decay component of the preceding response (a single exponential curve common under control and photoinhibited conditions). Here, we used the mean amplitude to avoid picking up spurious peaks in background noise and chose the time window (18–23 ms) to cover the peak of the photosensitive component in awake mice (Fig. 2e). Consequently, any change later than 23 ms from the onset of stimulation was ignored in this type of amplitude measurement. EPSCs and action potentials in whole-cell recordings were detected using TaroTools (https://sites.google.com/site/tarotoolsregister/), a threshold-based algorithm in Igor Pro (WaveMetrics). The event number (for both synaptic events and action potentials) evoked by stimulation was counted (baseline subtracted) in a time window adjusted for each cell to include all evoked events but to minimize contamination of spontaneous events. The beginning of the time window was adjusted to between 0 and 5 ms from the onset of stimulation, and the end of the time window was between 50 and 60 ms.

The border between the early and the late phases was adjusted to between 16 and 20 ms so that the border matched the valley of the time histogram. The synaptic charge was measured as the integral of the averaged current trace (baseline subtracted and sign reversed). The latency of the sensory response was defined as the time from the stimulus onset to the first EPSC event.

**Statistics and reproducibility**. Data are presented as means ± SEMs. Statistical significance was tested by using Wilcoxon signed-rank tests for paired samples and Wilcoxon–Mann–Whitney tests for two independent datasets. The significance of linear correlation was tested using Student's $t$ distribution. Nonparametric multiple comparison tests were performed using the PMCMR package in R (http://CRAN.R-project.org/). Kruskal–Wallis tests followed by pairwise tests for multiple comparisons of mean rank sums (Nemenyi test) were used for data in Fig. 5c and Supplementary Fig. 5b. Friedman tests followed by Nemenyi tests were used for data in Supplementary Fig. 5a. No statistical methods were used to predetermine sample sizes. Statistical tests were two-sided, and differences were considered statistically significant when $P$ values were <0.05.

**Reporting summary**. Further information on research design is available in the Nature Research Reporting Summary linked to this article.

## Data availability
The source data used to generate the charts present in the manuscript are provided in Supplementary Data 1. The data that support the findings in this study are available upon request from the corresponding author.

## Code availability
Code used for analysis is available at https://sites.google.com/site/tarotoolsregister/.

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

## Acknowledgements

The authors are grateful to Toshihiko Momiyama, Soichi Nagao, Hirofumi Tokuoka, Tadaharu Tsumoto and Kazuhiro Sohya for their generous support, to Paul Chadderton, Ian Duguid, Kouichi Hashimoto, and Yukihiro Nakamura for helpful comments on the

manuscript, and to Miwa Takagi for technical assistance. This work was supported by JSPS KAKENHI (18K06529, T.I. and 19K06919, I.S.), MEXT-Supported Program for the Strategic Research Foundation at Private Universities (S1311009, T.I.), and the Jikei University Research Fund (M.S. and T.I.).

## Author contributions

Conceptualization: M.S. and T.I., Methodology: M.S. and T.I., Software: T.I., Formal analysis: T.I., Investigation: M.S. and T.I., Resources: I.S., Writing–original draft: T.I.; Writing–review and editing: M.S. and I.S., Supervision: I.S. and T.I., Project administration: T.I., Funding acquisition: M.S., I.S., and T.I.

## Competing interests

The authors declare no competing interests.
