## [Peer Review File · Communications Biology]

Reviewers' comments:

Reviewer #1 (Remarks to the Author):

The major point of the paper is to show that there is convergent input into single cerebellar granule cells from brainstem and neocortex. Furthermore, these inputs are related and their afferent area in the neocortex is precisely defined.

These findings are novel and close a large gap in our knowledge. This convergence was suspected, but also disputed, and never shown functionally. In that sense this study is important. This makes the paper primarily of interest for cerebellar neuroscientists, but also for scientists investigating the broader movement system and cognitive systems, since the neocortex makes prominent connections to the cerebellum.

Specific remarks:

In the results section several latencies are reported which on first glance seem contradictory to the story, but they receive little attention in the manuscript, which might confuse readers or obfuscate interpretation of the data.

On lines 78 and 79 the latencies of responses in GCL and S1 in waking mice are listed, but the second peak response in GCL precedes the peak response in S1. This needs some additional clarification. Of course, the peak LFP does not equal the moment the signal leaves the cortex, but it would be good to explain this explicitly. The authors could consider adding the onset of the response. Certainly, the onset of the S1 response must precede the onset of the second GCL response. One could even consider that the response remaining in the field potential in S1 during illumination might represent the incoming synaptic signal. Comparing this latency might maybe show something?

Additionally, I would advice the authors to include two panels in figure 1 (there is space) that shows the latencies to peak or to onset for both anesthetized and waking experiments. This will help readers compare between the two conditions.

Another issue concerning the latencies is the window during which responses were compared between baseline and light stimulus. Because the timing of the peak shifts between KX and awake, the window for analysis shifts as well. But the choice of the extremely narrow window for the comparison of the late response in awake is not properly justified. It just so happens that the chosen window picks up the difference in the peak in line with the KX results, but ignores the more prominent difference occurring ~30-80ms, which is actually going the other way. This needs to be quantified and discussed at the very least.

I really like the experiments in figure 1f, which clearly shows that the location of the largest response in S1 correlates with the best location for inhibition. This is an important finding I think, but it is almost a bit lost now. It would be nice to make this more explicit. Is there a statistical test to formalize this a bit more? In other words, is there a statistical way to show that the two grids in 1f are not significantly different? This should be possible to show with a general linear model or maybe even with a variant on the X2 test.

There is a curious difference between the amplitude and area under the curve analyses. In figure 3e the amplitude is shown to be larger for early events compare with late events, but the area under the curve is larger for late events, as shown in suppl 3b. This is why the remark on lines 136-138 confuses me. Or does this remark only apply to the effect of the light? Maybe a little note can be added somewhere (maybe the legend to suppl 3?) explaining why the area under the curve differs from

amplitudes (I guess this is due to the decay time of the current?).

I'm missing some sort of small conclusion after line 143 where the Golgi cell inputs are discussed. It would be nice to know what the authors see in this data (e.g. why the 3.5ms delay between EPSC and IPSC, did the authors also see late (putative) Golgi cell-mediated responses, during and right after the 1 second illumination, was there a shift in the holding potential at 0mV as would be expected due to decreased (tonic) inhibition?)

The difference in resting-membrane potentials is curious since most other reports don't report this (e.g. Powell 2015, Ishikawa 2015). This is potentially worrying since these differences might be explained (in part) by leaky recording conditions, and since the authors don't report the seal resistance obtained before breaking into whole cell mode. It would be good for the authors to critically assess what might explain this difference in holding. A first indication might be provided by looking at the apparent membrane resistance since a leaky seal might artificially lower the apparent membrane resistance. Offset problems might be detected by looking at the firing thresholds of the neurons, these should be in a very confined voltage range of only a few mV.

The residual voltage after the first burst of inputs is described in fig 4e. The attempt to explain this is commendable, but there seem to be fundamental differences between figure 4e and supplementary figure 4. First and foremost, in the data in 4e it seems that the residual is many times larger than in the model in supplementary figure 4. Second, when comparing figures 4e and 4d it seems that the four neurons with the largest residual are not very prominent late spikers. This leads to the conclusion that many other mechanisms are in play.

Therefore I think that the authors are right in being careful in interpreting these results. But then I think we should be honest and not try to interpret this data at all. The model does not add anything at this point and there is basically nothing here. The best explanation is that the second burst of APs is there because of the second burst of inputs, but the correlation between events and APs is weak because of residual, feedback and feedforward inhibition, etc. The discussion should be adapted as well in this case.

The analysis done for figure 5d is really strong. Why was this analysis not done for the neurons recorded for figure 3?

The discussion on whether corticopontine fibers are located differently in the granule cell layer is interesting, but lacks some detail. For example Quy et al (J Comp Neurol, 2011) shows in their figure 12 that corticopontine fibers are indeed located higher in the granule cell layer. But there are also other organizations such as those described by Ekerot & Larson (Brain Res, 1972) where primary afferents have different distributions over the lobule. With respect to the overlap, considering the results from Quy et al, this would not be unexpected. There does seem to be a large overlap.

The presence of a pathway that modulates complex spike responses in the cerebellar cortex from the sensory-motor cortex has been described before. Schwarz and Welsh (J Neurophys, 2001) show that stimulation of M1 modulates simple and complex spike activity in Purkinje cells. In this light it is interesting to discuss the trials with and without CS both in your experiments and those in Schwarz and Welsh.

General and small remarks:

Correlations are all given without their significance. Please correct to show which correlations are significantly different from zero.

Line 131: reference to 3c, should be 3d?

Line 139: reference to 3cd, should be 3fg?

Line 143: reference to 3e-g, should be cd?

Reviewer #2 (Remarks to the Author):

COMMSBIO-19-1906-T

Shimuta et al. examine the integration of sensory information by the cerebellum, and utilize in vivo electrophysiology and optogenetic neural manipulation to determine how direct and indirect sensory input are processed within the cerebellar cortex. This manuscript presents three important findings in a clear and coherent manner: 1) direct (from trigeminal nuclei) and indirect (cortico-ponto-cerebellar pathway) sensory information converge onto the same granule cells, 2) integration of sensory information influences not only granule cell output but also Purkinje cell output, 3) climbing fiber input to Purkinje cells, an important locus of plasticity, depends on the activity of sensory cortical input. The rationale and justification for the experiments as well as the experimental procedure, results and interpretations are clearly described. The findings described in this manuscript complement, and nicely extend findings by two earlier studies (Huang et al., 2013; Ishikawa et al., 2015). The only concern is that a few statements are made based on low n numbers in (see below). Major and minor comments and suggestions for the manuscript are outlined below.

Major comments:

1. For Supp. Figure 1, please clarify in the results section whether illumination pattern G is statistically different from control for the late phase of GCL response, since illumination pattern G inhibits S1 prior to air puff.
2. In awake animals, GCL responded to airpuff with 3 peaks (Figure 2c). Please explain how the first 2 peaks are determined to be early component, and third peak as late component in the GCL response of anesthetized mice (page 3).
3. Please describe potential pathways in addition to the two tested by the authors. This is especially important since awake mice have early, intermediate and late peaks, as opposed to 2 phases in the anaesthetized mice (Fig. 1a-d).
4. For Figure 3a, the representative GCL in anaesthetized mouse exhibits 3 peaks, similar to an awake mouse, but different from the anaesthetized mouse shown in Figure 2. Please explain this discrepancy.
5. Please comment on the variability in GCL response between the 24 cells. Does this reflect a technical challenge in recording from the same location or were there slight changes in coordinates? Any correlation between distinctions in response and how different populations of GCL were targeted might be informative.
6. For Supp. Figure 3b, majority of late phase controls have very low synaptic charge to begin with. Only about 7 of 24 cells presented have synaptic charges of more than 200 pA*ms with a significant decrease after photoinhibition. Please clarify whether neurons without late phase peak are included in this data.

7. For Figures 3e-g, 5c and Supp. Figures 3c-d, 5a-b, the n numbers are quite low. Please increase n number for statistical sensitivity. For instance, the statements about differential mossy fiber inputs across parasagittal bands are based on very low n numbers.

Minor comments:

1. Please provide references for page 2, statement on lines 26-27, statement on lines 36-38, and statement on lines 48-49.

2. Please provide stereotaxic coordinates for other loci ("distant" ones) in methods or in supplemental results (page 4, line 90).

3. For Figure 2f, please clarify "amplitude of S1 field potential" and "%inhibition of cerebellar late component" across different coordinates.

4. Page 5, lines 129-131, Jitter events refer to Fig. 3d, not Fig. 3c.

5. The n numbers for Fig. 5a are not reported. Are these 5 representative traces from each population or were more recorded?

6. Page 9, Fig. 5b (line 203) is introduced before Fig. 5a (line 206).

7. For Supp. Fig. 4, please clarify in the text how the computer simulator supports the result in Fig. 4.

8. For Fig. 6a, please include missing n numbers for PCs and trials.

9. Page 15, line 344, this may be possible, in principle, (commas around in principle).

Reviewer #3 (Remarks to the Author):

In this manuscript, authors explore the way in which sensory information is processed in the cerebellum by looking at electrophysiological properties of different cell types in the cerebellum, specifically the granule cell and purkinje cell populations. Marr and Albus speculated that granule cells receive mossy fiber inputs that originate from different areas of the brain (trigeminal/corticopontine); some granule cells receiving from a single area, others from multiple. Recent studies have confirmed this but what is unclear in the literature and novel in this study is how these different types of mossy fiber inputs converge on granule cells; their results showing that sometimes a granule cell has a single input type, sometimes multiple. The authors speculate that this array of input integrates the peripheral and cortical signals, the sum of this producing the final output to the purkinje neurons. To prove this, the authors use optogenetics to ultimately inhibit a specific region of the neocortex where sensory information is relayed (done by stimulating rhodopsin+ interneurons which then in turn inhibit the neurons which project to the corticopontine). They then examine how inhibition of this one aspect (via corticopontine) but leaving the other intact (trigeminal) modulates electrical activity of first, the granule cell neurons and in turn the purkinje neurons. Interestingly, both field potential and patch clamp single cell recording of this setup produces an early and late electrical response to tactile stimulation (air puff); authors deducing that early is in response to direct trigeminal pathway and late due to multi-step corticopontine pathway (through cortex). Through a series of experiments, they show meticulously and convincingly how inhibition of the corticopontine pathways can affect both electrophysiological input and output of the granule cells. They then demonstrate how this will then

alter the simple spiking of purkinje neurons. Intriguingly and greatly adding to novelty of the study, they further show through this optogenetic inhibition model that tactile stimulation via S1 further affects purkinje neurons by inducing complex spiking via climbing fibers.

In general, this study is well designed, carefully performed, and all aspects well addressed in either the text or in supplemental experiments. The results are in my opinion convincing and intriguing. Although previous studies describing the different mossy fiber pathways detracts from the novelty of this study, the authors meticulous untangling of the relationship with these pathways and granule cells is exciting and novel. Furthermore, demonstration that inhibition of S1 affects climbing fiber mediated complex spiking is a novel finding.

One major criticism of this study is that the majority of studies have been performed in anaesthetized mice whereas the most convincing study would be in awake, drug naïve mice. A further criticism on this point is that different anesthetics were tested and it could seem as the one with the best results was chosen (though on a positive showing these results demonstrates transparency). The authors do however, give good reasoning for their decision with their final setup. To increase the impact of this paper it would have been pertinent to also demonstrate the inhibition of the trigeminal pathway and test the cell dynamics this would produce. The authors do however recognize this short-coming and provide adequate explanation as to why they did not pursue this.

The statistical analysis appear to be appropriate and robust.

Minor points:

I do not find the second peak in awake mice (figure 2) convincing. It looks as if there is perhaps no effect of the light. I suggest the authors further explain why there are 3 peaks in the awake mice (2 early peaks).

Why is there such an effect of the light in the S1 field in awake vs K/X (Fig 2a vs c). This should be addressed in the text.

In general, the manuscript text itself is very well written and free of confusion. I would suggest the following minor adjustments to the flow:

Figure 2 – when talking about early and late peaks, it is difficult at the beginning to know which they are referring to. The colored bars underneath do not do well enough a job to guide readers to understand which peak is being talked about. I suggest a schematic drawing with numbers or labels on the referenced peaks.

Figure 2. Make clear in title of figure that it is field potentials. Few clues in the figure text as to what type of recordings is (ie cell vs field)

Fig. 6a. The example cell is not particularly good at demonstrating the early burst in activity.

Point-by-point response to reviewers:

Reviewer #1:

1: In the results section several latencies are reported which on first glance seem contradictory to the story, but they receive little attention in the manuscript, which might confuse readers or obfuscate interpretation of the data. On lines 78 and 79 the latencies of responses in GCL and S1 in waking mice are listed, but the second peak response in GCL precedes the peak response in S1. This needs some additional clarification. Of course, the peak LFP does not equal the moment the signal leaves the cortex, but it would be good to explain this explicitly. The authors could consider adding the onset of the response. Certainly, the onset of the S1 response must precede the onset of the second GCL response. One could even consider that the response remaining in the field potential in S1 during illumination might represent the incoming synaptic signal. Comparing this latency might maybe show something? Additionally, I would advice the authors to include two panels in figure 1 (there is space) that shows the latencies to peak or to onset for both anesthetized and waking experiments. This will help readers compare between the two conditions.

Response: We apologize that the data in our initial manuscript were not described clearly enough to avoid confusion. We have now clarified that the GCL response had three peaks. The second peak to which the reviewer refers to preceded the peak of S1 and was independent of S1 activity. By contrast, the third peak was S1 dependent. We have rewritten the description in the text (beginning on line 86) and added a figure panel to show that neither the first or second peak is affected by S1 inhibition (Figure 2d). Furthermore, as the reviewer suggested, we added two panels to visualize the latencies to peaks (Figure 2b and d, top right).

2: Another issue concerning the latencies is the window during which responses were compared between baseline and light stimulus. Because the timing of the peak shifts between KX and awake, the window for analysis shifts as well. But the choice of the extremely narrow window for the comparison of the late response in awake is not properly justified. It just so happens that the chosen window picks up the difference in the peak in line with the KX results, but ignores the more prominent difference occurring ~30-80ms, which is actually going the other way. This needs to be quantified and discussed at the very least.

Response: We agree that there is a risk of detecting a spurious peak in this type of analysis. Therefore, we took extra care in this analysis. First, we fixed the time window across all cases to avoid subjective case-by-case adjustments. Second, we used the mean amplitude (i.e., the average of all points in the window) rather than the peak amplitude within a window because the mean amplitude is less affected by background noise. We subtracted the slope component by curve fitting, but this process subtracted the same value from the control and the photoinhibited conditions. This analysis required a relatively narrow window (5 ms duration) covering the peak of the response. We added an explanation of this analysis in Methods (line 474).

As the reviewer pointed out, there was a change in the opposite direction after 30 ms. This upward component appeared to be a rebound after the downward response (Figure 2e), but it was not consistently observed across all cases. However, we would like to avoid describing details about this component for two reasons. First, no matter what we find about this component, it will not add a logical justification to the setting of the window discussed above because they are different issues. Second, under our recording conditions, slow waves were more difficult to analyze than fast waves. As we can see in the sample recordings in Figure 2C, there was slow fluctuation in the waveform even before applying a stimulation. Such background fluctuation can easily affect the measurement of small slow changes. Therefore, we would like to refrain from providing potentially misleading information.

3: I really like the experiments in figure 1f, which clearly shows that the location of the largest response in S1 correlates with the best location for inhibition. This is an important finding I think, but it is almost a bit lost now. It would be nice to make this more explicit. Is there a statistical test to formalize this a bit more? In other words, is there a statistical way to show that the two grids in 1f are

not significantly different? This should be possible to show with a general linear model or maybe even with a variant on the X2 test.

Response: We thank the reviewer for this suggestion. We calculated the correlation coefficient between the size of the S1 response and the inhibition rate across 29 spots. The correlation was highly significant ($P < 0.001$) and was added to the legend for Figure 2. We conducted the same analysis for data in Figure 6e, which also showed a statistically significant difference, as described in the legend for Figure 6.

4: There is a curious difference between the amplitude and area under the curve analyses. In figure 3e the amplitude is shown to be larger for early events compare with late events, but the area under the curve is larger for late events, as shown in suppl 3b. This is why the remark on lines 136-138 confuses me. Or does this remark only apply to the effect of the light? Maybe a little note can be added somewhere (maybe the legend to suppl 3?) explaining why the area under the curve differs from amplitudes (I guess this is due to the decay time of the current?).

Response: The amplitude presented in Figure 3e is the amplitude of individual EPSC events. Therefore, the area under the curve (Supplementary Figure 3b) is likely to correspond to the amplitude (Figure 3e) multiplied by the number of events/trial (Figure 3f). In this regard, we interpreted that the area under the curve was larger for the late components because they had more events. It is also possible that, as the reviewer suggested, there may be some difference in the waveform of individual EPSCs. However, it is very difficult to quantify the decay time in our data because most of the events overlapped.

5: I'm missing some sort of small conclusion after line 143 where the Golgi cell inputs are discussed. It would be nice to know what the authors see in this data (e.g. why the 3.5ms delay between EPSC and IPSC, did the authors also see late (putative) Golgi cell-mediated responses, during and right after the 1 second illumination, was there a shift in the holding potential at 0mV as would be expected due to decreased (tonic) inhibition?)

Response: We mentioned these results in Discussion (Line 337). Basically, Golgi cells receive the same mossy fiber inputs as neighboring granule cells. Golgi cells then send rapid feed-forward inhibition to the granule cells. Therefore, in the granule cells, IPSCs comes shortly after EPSCs with a delay of monosynaptic transmission (3.5 ms). Since there are many papers describing the Golgi-granule cell connection, including *in vivo* recordings from Golgi cells [ref. 57], we only briefly discussed this issue. As the reviewer raised, it may be interesting to see whether S1 inhibition affects Golgi cell-mediated tonic inhibition. However, we cannot answer this question because we tested only brief (150 ms) cortical inhibition while recording IPSCs in granule cells.

6: The difference in resting-membrane potentials is curious since most other reports don't report this (e.g. Powell 2015, Ishikawa 2015). This is potentially worrying since these differences might be explained (in part) by leaky recording conditions, and since the authors don't report the sealresistance obtained before breaking into whole cell mode. It would be good for the authors to critically assess what might explain this difference in holding. A first indication might be provided by looking at the apparent membrane resistance since a leaky seal might artificially lower the apparent membrane resistance. Offset problems might be detected by looking at the firing thresholds of the neurons, these should be in a very confined voltage range of only a few mV.

Response: The variety of resting membrane potentials of granule cells was originally reported by Chadderton et al. (Nature, 2004, ref 28) for anesthetized rats. Their histogram (Figure 1b of their paper) ranged from -86 to -38 mV, and the mean was -64 mV. Although their standard deviation was not reported, it is estimated to be around 10 mV from the histogram. Powell et al. (eLife, 2015, ref 75) reported potentials of -67.0 ± 8.9 mV (mean \pm SD) for awake mice. The value in our present paper was -72.9 ± 13.8 mV (mean \pm SD) (SEM = 3.4 mV, $n = 16$) for anesthetized mice, which does not differ substantially from the previous reports in the sense that there is a wide variety of resting potentials. Still, it is not clear why our resting potential was relatively hyperpolarized. The quality of our recordings was good, having high seal resistance (>5 G Ω) and high whole-cell membrane resistance

(1.57 ± 0.16 G Ω ; mean \pm SEM, $n = 25$). We added the membrane resistance value to the Methods section (line 449). We also confirmed after each recording that the voltage drift of the reference electrode was small (<5 mV). As the reviewer suggested, we looked into the threshold potential. Except for a few silent cells that never fired, the firing threshold was confined to -46.5 ± 5.8 mV (mean \pm SD, $n = 14$) and was not correlated to the resting potential ($P = 0.88$). Taken together, we believe that the wide variation of the resting potentials was genuine.

7: The residual voltage after the first burst of inputs is described in fig 4e. The attempt to explain this is commendable, but there seem to be fundamental differences between figure 4e and supplementary figure 4. First and foremost, in the data in 4e it seems that the residual is many times larger than in the model in supplementary figure 4. Second, when comparing figures 4e and 4d it seems that the four neurons with the largest residual are not very prominent late spikers. This leads to the conclusion that many other mechanisms are in play.

Therefore, I think that the authors are right in being careful in interpreting these results. But then I think we should be honest and not try to interpret this data at all. The model does not add anything at this point and there is basically nothing here. The best explanation is that the second burst of APs is there because of the second burst of inputs, but the correlation between events and APs is weak because of residual, feedback and feedforward inhibition, etc. The discussion should be adapted as well in this case.

Response: We agree that the benefit of the computational models is limited. However, we respectfully disagree with the reviewer on the first point because the residual voltages in Figure 4e are similar to or smaller than (but not “many times larger than”) those in the model in Supplementary Figure 4, where the residual voltages are 6.3, 7.1, 7.8, and 7.8 mV in panel a1, a2, a3, and a4, respectively. Thus, the model matches the real data in this aspect. The second point the reviewer mentions is that the neurons with large residual voltages were not necessarily late spikers (because these cells have other reasons not to spike). This is the case, and we carefully and honestly interpreted the results in the previous version of the manuscript to conclude that “direct quantification of such a facilitating effect was not feasible in our experimental data because of involvement of many other factors”. However, we admit the subsequent sentence about the computational model might have been misleading. Thus, we have amended the sentence (from line 191) to “Although direct quantification of such a facilitating effect was not feasible with our experimental data because of the involvement of many other factors (including the probability of firing in the early phase and the size of synaptic inputs in the late phase), a simulation with a realistic computational model of a granule cell³³ demonstrated such a facilitation by temporal summation (Supplementary Figure 4)”, using the word “demonstrated” instead of “reproduced” because this model is somehow independent from the experimental data.

We agree that it is not easy to interpret our results. However, we prefer to disclose our interpretation because we applied a bottom-up approach as well as a top-down approach, to some extent, to interpret the experimental results. The basics of neurophysiology state that an EPSP facilitates subsequent firing. However, an EPSP may have the opposite effect in some cases because (1) depolarization reduces the driving force of excitatory AMPA and NMDA currents, (2) depolarization increases the driving force of inhibitory GABAergic current, (3) a prolonged depolarization inactivates Na⁺ channels, and (4) Ca²⁺ influx via NMDARs activates Ca²⁺-activated K⁺ channels, which increase the dendritic leak current. All these factors are incorporated in the computational model in Supplementary Figure 4 (originally by Diwakar et al., ref 33), and the simulation result shows that the residual depolarization facilitates subsequent firing in a model of cerebellar granule cells. This is not a big surprise but meaningful support for our interpretation. We added an explanation to the legend of Supplementary Figure 4.

8: The analysis done for figure 5d is really strong. Why was this analysis not done for the neurons recorded for figure3?

Response: We took an observational approach for the data in Figure 5d and an interventional approach for those in Figure 3 to prove the same principle. We did not calculate correlation coefficients in Figure 3 because the result was evident from the effect of the optogenetic intervention.

9: The discussion on whether corticopontine fibers are located differently in the granule cell layer is interesting, but lacks some detail. For example, Quy et al (J Comp Neurol, 2011) shows in their figure 12 that corticopontine fibers are indeed located higher in the granule cell layer. But there are also other organizations such as those described by Ekerot & Larson (Brain Res, 1972) where primary afferents have different distributions over the lobule. With respect to the overlap, considering the results from Quy et al, this would not be unexpected. There does seem to be a large overlap.

Response: We thank the reviewer for this suggestion. We have cited those papers (ref. 50 and 52) in the paragraph starting at Line 330 in the Discussion.

10: The presence of a pathway that modulates complex spike responses in the cerebellar cortex from the sensory-motor cortex has been described before. Schwarz and Welsh (J Neurophys, 2001) show that stimulation of M1 modulates simple and complex spike activity in Purkinje cells. In this light it is interesting to discuss the trials with and without CS both in your experiments and those in Schwarz and Welsh.

Response: We thank the reviewer for directing our attention to this paper by Schwarz and Welsh. However, it is difficult for us to compare their results with ours because there is a discrepancy of basic ideas. While they thought that the pause of SS after M1 stimulation was caused by climbing fiber activation even in the absence of CS, we think that the pause was caused by the activation of molecular layer interneurons (MLI), which were activated by parallel fibers. Perhaps they assumed that MLI are hardly activated when SS are absent, but this assumption is not supported by our present knowledge on the cerebellar circuit.

11: Correlations are all given without their significance. Please correct to show which correlations are significantly different from zero.

Response: As requested, we added the results of statistical tests at line 136, 188, and 189.

12-14:

Line 131: reference to 3c, should be 3d?

Line 139: reference to 3cd, should be 3fg?

Line 143: reference to 3e-g, should be cd?

Response: We thank the reviewer for pointing out these. We corrected the errors at lines 145, 153 and 157.

Reviewer #2:

Major comments:

1. For Supp. Figure 1, please clarify in the results section whether illumination pattern G is statistically different from control for the late phase of GCL response, since illumination pattern G inhibits S1 prior to air puff.

Response: We increased the sample size from 5 to 8 in this experiment, and it was possible for us to apply statistical tests. However, we are afraid that we may confuse readers if we use statistical tests here because statistical “non-significance” does not mean that there was no effect. In this experiment, we intended to show that not only pattern C but also some other patterns were effective to suppress the late component of GCL response. We believe that our result is clear without statistical tests. Pattern G may be interesting in itself because it is on the borderline of effectiveness. However, we would like to avoid describing this pattern in detail because we did not use it in the rest of this paper. For accuracy, we have reworded the text to state that the illumination “covering the timing of the sensory stimulation” was effective in Results (line 85).

2. In awake animals, GCL responded to airpuff with 3 peaks (Figure 2c). Please explain how the first 2 peaks are determined to be early component, and third peak as late component in the GCL response of anesthetized mice (page 3).

Response: This point is related to the 1st point raised by Reviewer #1. In the previous version of the manuscript, we called the first two peaks “early” because their timing corresponded to the early component in the K/X condition, and in the same sense, we called the third peak “late”. However, we now refer to them as the 1st, 2nd, and 3rd peaks to avoid confusion. We have reworded the text (line 86) and added figure panels (Figure 2b,d).

3. Please describe potential pathways in addition to the two tested by the authors. This is especially important since awake mice have early, intermediate and late peaks, as opposed to 2 phases in the anaesthetized mice (Fig. 1a-d).

Response: We thank the reviewer for this suggestion. We have added the following description to the Discussion (line 374): “It is possible that rapid polysynaptic inputs contributed, such as those relayed within the trigeminal nuclei or in other brain areas, including the pontine nuclei, which send mossy fibers to the cerebellar cortex⁶⁷, and the cerebellar nuclei, which project via nucleo-cortical mossy fibers⁶⁸.”

4. For Figure 3a, the representative GCL in anaesthetized mouse exhibits 3 peaks, similar to an awake mouse, but different from the anaesthetized mouse shown in Figure 2. Please explain this discrepancy.

Response: The cell in Figure 3a has two early peaks in the bins from 6 to 8 ms and from 10 to 12 ms. Both of these bins correspond to the early component in anesthetized mice in Figure 2a-c. At the single-cell level, we often saw these peaks within the early component because the first few EPSCs are well time locked. The cell in Figure 5a has a similar histogram. However, in the field potential recordings, which reflect the summation of signals from many cells, these peaks are smeared and we see only one peak. We hope that the panels about peak times that we added (Figure 2b and d, top right) will help readers to understand.

5. Please comment on the variability in GCL response between the 24 cells. Does this reflect a technical challenge in recording from the same location or were there slight changes in coordinates? Any correlation between distinctions in response and how different populations of GCL were targeted might be informative.

Response: We added the following sentence to the Result section before describing the data from 24 cells: “Granule cells in the central part of crus II (corresponding to 5-, 6+, and 6- bands) were sampled without visual inspection.” (line 129).

As our “blind” patch-clamp method lacks morphological information at the microscopic level, it is difficult to directly answer the reviewer’s question regarding the variability of responses and the locations of the cells. Perhaps we need imaging methods to answer the question, as we stated in Discussion (line 335). In light of this comment from the reviewer, we amended our discussion about the variability of inputs in the paragraph beginning line 321. We omitted mentions of “by chance” and “randomness” because these words may be misleading given that we do not know whether the cellular organization was randomly formed or only our sampling was random.

6. For Supp. Figure 3b, majority of late phase controls have very low synaptic charge to begin with. Only about 7 of 24 cells presented have synaptic charges of more than 200 pA*ms with a significant decrease after photoinhibition. Please clarify whether neurons without late phase peak are included in this data.

Response: We added a description to the legend stating that all cells, including those without the late component, were included in this analysis. This may dilute the effect of photoinhibition but the effect was still highly significant ($P < 0.001$).

7. For Figures 3e-g, 5c and Supp. Figures 3c-d, 5a-b, the *n* numbers are quite low. Please increase *n* number for statistical sensitivity. For instance, the statements about differential mossy fiber inputs across parasagittal bands are based on very low *n* numbers.

Response: We performed additional experiments and added the *n* values to these figures. The reviewer wrote "Figure 3e-g", but Figure 3e,f has many cells and Figure 3 does not have panel g. Therefore, we supposed that the reviewer was referring to Supplementary Figure 3f,g. We have added two cells to the data in these panels.

For Figure 5c, the *n* value was increased to 7–9 (from 5–7) for each parasagittal band, and a statistical test showed a significant difference in the number of EPSC events for the late components between the 5- and 7+ bands. The same data set is presented in Supplementary Figures 5b–e. Supplementary Figure 3c–e has six cells (increased from four), and the statistical test showed significant suppression of the late component by S1 inhibition. Supplementary Figure 5a has seven animals (increased from five), and the statistical test showed a significant difference in the balance of the early and late components.

Minor points:

1. Please provide references for page 2, statement on lines 26-27, statement on lines 36-38, and statement on lines 48-49.

Response: We have provided the references (ref. 5 and 13) as requested.

2. Please provide stereotaxic coordinates for other loci ("distant" ones) in methods or in supplemental results (page 4, line 90).

Response: We wrote the coordinates in Methods (line 463).

3. For Figure 2f, please clarify "amplitude of S1 field potential" and "%inhibition of cerebellar late component" across different coordinates.

Response: We made it clear that the bubble size is proportional to each index and added labels to show one more value (in M1) in each panel so that readers can easily recognize the values.

4. Page 5, lines 129-131, Jitter events refer to Fig. 3d, not Fig. 3c.

Response: We have corrected this error.

5. The *n* numbers for Fig. 5a are not reported. Are these 5 representative traces from each population or were more recorded?

Response: They are representative traces, showing five repeated trials from each cell. Pooled data are shown in Figure 5c–e and Supplementary Figure 5b–e. We have added a sentence in the legend of Figure 5 to make this clear.

6. Page 9, Fig. 5b (line 203) is introduced before Fig. 5a (line 206).

Response: We will discuss this issue with the editors.

7. For Supp. Fig. 4, please clarify in the text how the computer simulator supports the result in Fig. 4.

Response: This is related to Reviewer #1's point 7. We have added an explanation to the legend of Supplementary Figure 4.

8. For Fig. 6a, please include missing *n* numbers for PCs and trials.

Response: Figure 6a shows 40 representative trials from a Purkinje cell. We have added this number to the legend. Pooled data (*n* = 20 Purkinje cells) were used for Figure 6c,d.

9. Page 15, line 344, this may be possible, in principle, (commas around in principle).

Response: We have added commas as requested.

Reviewer #3:

Minor points:

I do not find the second peak in awake mice (figure 2) convincing. It looks as if there is perhaps no effect of the light. I suggest the authors further explain why there are 3 peaks in the awake mice (2 early peaks).

Response: This is related to the 1st and 2nd comments from Reviewer #1, and the 2nd major comment from Reviewer #2. We have amended this part of the Results (starting from line 86) to provide a more straightforward description, and we added more information to the Methods (line 478) to explain the extra care we took to avoid biasing the analysis. We do not know why there is a 2nd peak in the awake mice, but we discuss that this may reflect a polysynaptic component in “Limitations and future directions” in the Discussion (from line 372).

Why is there such an effect of the light in the S1 field in awake vs K/X (Fig 2a vs c). This should be addressed in the text.

Response: The reviewer is pointing out that the light shifted the baseline downward in the K/X condition but upward in the awake condition. This difference was consistently observed in different mice, but we cannot explain why there was such a difference. This may be related to the excitatory/inhibitory balance in the neocortex in different brain states caused by anesthetics. As we cannot give a concrete explanation and this is a side issue, we would like to refrain from mentioning this issue in the text.

In general, the manuscript text itself is very well written and free of confusion. I would suggest the following minor adjustments to the flow:

Figure 2 – when talking about early and late peaks, it is difficult at the beginning to know which they are referring to. The colored bars underneath do not do well enough a job to guide readers to understand which peak is being talked about. I suggest a schematic drawing with numbers or labels on the referenced peaks.

Response: We thank the reviewer for this suggestion. We added labels to the bars in the graphs in Figure 2.

Figure 2. Make clear in title of figure that it is field potentials. Few clues in the figure text as to what type of recordings is (ie cell vs field)

Response: We added this information to the title of Figure 2.

Fig. 6a. The example cell is not particularly good at demonstrating the early burst in activity.

Response: The reviewer is pointing out that the cell shown in Figure 6a has only a small “early excitation” response. However, the responses of cells were variable; thus the pooled data in Figure 6c are best suited for readers to grasp the whole picture.

REVIEWERS' COMMENTS:

Reviewer #1 (Remarks to the Author):

Numbers refer to the numbering the authors applied in their response.

1. Much clearer now. I also now see where my confusion came from.
2. Maybe the slower late component is a different issue, maybe it is not, there is no evidence at the moment one way or the other. I find it hard to imagine this this late component is some form rebound, since seems to be absent or at least much faster in the 'non-light' trace in 2C. If this late slow component is indeed very variable then I might agree with the authors that this is the best possible way to work with the data: setting the window narrow and consistently over all cases to analyse only the fast peak. But, please be explicit in the M&M about this.
3. Wonderful! Happy to contribute some suggestions this way.
4. I now see where these differences come from. I would suggest though to make more explicit in the text how this should be interpreted. I would suggest a more elaborate legend for suppl. 3.
5. First, I think my comments were not worded entirely clearly: Wouldn't you also expect a change in IPSC rate later on reflecting feedback inhibition (MF-GrC-GoC-GrC)?
The authors state that the 3.5ms extra delay reflects the one extra monosynaptic delay (MF-GoC-GrC). But why not state that explicitly in the text? "...had a 3.5 ms difference, reflecting one additional monosynaptic delay from MF-GoC-GrC" or something alike.
It is a shame that the authors did not test longer illuminations when recordings IPSCs (like in suppl 3F). But, I won't ask for this since it's a small thing.
6. Thank you, I am fully reassured. Great to see the care in this.
7. This was badly worded on my part, apologies. When I look at figure 4 I see ~2-3mV depolarization. But I think suppl 4 shows up to more than double that.
Irrespectively, we can work with this if the text is changed.
In 191: "Although direct quantification of such a facilitating effect was not feasible with our experimental data" I think this should be edited to mean explanation, not quantification since the modeling data is hard to interpret quantitatively. Thus, this last section of the paragraph creates false expectations.
If the aim of the model is, as the authors now seem to suggest in their response to the reviewers, that it is to show that even when counteracting ephys sources are incorporated, depolarization remains a likely mechanism, state that in the last paragraph.
8. I think any evidence that would make the paper or the arguments in the paper stronger should be taken. But it's your call to add this or not.
9. Happy to suggest!
10. I can see this argument, agreed.
- 11-14. Thank you.

In general many of my initial remarks have been fully resolved. I think the text around the modeling can be improved to avoid creating false expectations. I still have my doubts about the utility of the model, but I can agree with a carefully worded text around the model.

Reviewer #2 (Remarks to the Author):

COMMSBIO-19-1906-T

Comments for revised manuscript:

We are satisfied with how the authors have addressed most of our questions and concerns in this

revised manuscript, and congratulate the authors on a very nice study.

Reviewer #3 (Remarks to the Author):

After reviewing the changes made as a result of our comments to the manuscript entitled "Convergence of unisensory-evoked signals via multiple pathways to the cerebellum" by Dr Ishikawa and colleagues we are satisfied and would recommend for publication. Specifically our major point of contention was also brought up by other reviewers and has subsequently been firmly addressed by the authors. Minor points were also addressed adequately and we are satisfied with the authors' wish to refrain from addressing the anomaly brought up in our second comment. The final manuscript reads very well and we look forward to it being published.

Point-by-point response

From the editor:

1. Specifically, based on the recommendation from reviewer #1 we ask that you please explicit state in the manuscript how was the GCL late component determine and why. Please also include any potential limitations of this analysis.

Response: In the Methods section (Line 411), we added explicit statement that the time window was set on the peak of the photosensitive component and that this setting had a limitation that any change after this time window was ignored.

“Here, we used the mean amplitude to avoid picking up spurious peaks in background noise and chose the time window (18-23 ms) to cover the peak of the photosensitive component in awake mice (Figure 2e). Consequently, any change later than 23 ms from the onset of stimulation was ignored in this type of amplitude measurement. “

2. We also ask that you please explicitly state in the figure legend of supplementary figure 3b that there are 17 cell of low charge, a concern raised by reviewer #2 in the last round, and that it is possible these might represent a different population of cells.

Response: Please note that we did not do anything special in this figure as we always included all cells. But, to make this clear, we added a explicit statement in the figure legend of supplementary figure 3b, **“All types of granule cells, including cells lacking a late component, were included.”**

From Reviewer #1:

2. Maybe the slower late component is a different issue, maybe it is not, there is no evidence at the moment one way or the other. I find it hard to imagine this this late component is some form rebound, since seems to be absent or at least much faster in the 'non-light' trace in 2C. If this late slow component is indeed very variable then I might agree with the authors that this is the best possible way to work with the data: setting the window narrow and consistently over all cases to analyse only the fast peak. But, please be explicit in the M&M about this.

Response: This point is related to the 1st point raised by Reviewer #1. In the last rebuttal, we wrote the slower (upward) late component was “a different issue” because it had to be dealt as an independent issue in a statistical sense. We agree with the reviewer that this component may be related to the faster (downward) late component in a biological sense and may be important in itself. However, we will need an additional series of experiments to understand what the slower (upward) late component is. In the present paper, we would like to focus on the faster (downward) late component. We added the explicit statement in the Method as described above.

4. I now see where these differences come from. I would suggest though to make more explicit in the text how this should be interpreted. I would suggest a more elaborate legend for suppl. 3.

Response: We thank the reviewer for this suggestion. We added a sentence in the legend for Supplementary Fig. 3, **“The synaptic charge, which can be measured without event detection, is an index related to both the number of events (Figure 3f) and the size of individual EPSCs (Figure 3e).”**

5. First, I think my comments were not worded entirely clearly: Wouldn't you also expect a change in IPSC rate later on reflecting feedback inhibition (MF-GrC-GoC-GrC)? The authors state that the 3.5ms extra delay reflects the one extra monosynaptic delay (MF-GoC-GrC). But why not state that explicitly in the text? "...had a 3.5 ms difference, reflecting one additional monosynaptic delay from MF-GoC-GrC" or something alike. It is a shame that the authors did not test longer illuminations when recordings IPSCs (like in suppl 3F). But, I won't ask for this since it's a small thing.

Response: We again thank the reviewer for the suggestion. We added in the legend for Supplementary Fig. 3, **“The onset (20% rising point) of the averaged traces had a 3.5 ms difference, reflecting one additional synaptic delay in the mossy fiber-Golgi cell-granule cell pathway.”**

7. This was badly worded on my part, apologies. When I look at figure 4 I see ~2-3mV depolarization. But I think suppl 4 shows up to more than double that. Irrespectively, we can work with this if the text is changed. In 191: "Although direct quantification of such a facilitating effect was not feasible with our experimental data" I think this should be edited to mean explanation, not quantification since the modeling data is hard to interpret quantitatively. Thus, this last section of the paragraph creates false expectations. If the aim of the model is, as the authors now seem to suggest in their response to the reviewers, that it is to show that even when counteracting ephys sources are incorporated, depolarization remains a likely mechanism, state that in the last paragraph.

Response: Following the reviewers advice not to use “quantification”, we made a straightforward statement **“we could not directly extract such a facilitating effect from our experimental data”** (Line 158).

We also explained the intention of the computational model in the last paragraph: **“a simulation with a realistic computational model of a granule cell demonstrated facilitation of firing in the late phase by temporal summation even when counteracting factors were included in the model.”** (Line 161)